# Validity and intrarater reliability of a novel device for assessing Plantar flexor strength

Seth O'Neill[1]*, Alice Weeks[1], Jens Eg Nørgaard[2,3], Martin Gronbech Jorgensen[2,3]

**1** School of Healthcare, College of Life sciences, University of Leicester, Leicester, United Kingdom,
**2** Department of Geriatric Medicine, Aalborg University Hospital, Aalborg, Denmark, **3** Department of Clinical Medicine, Aalborg University, Aalborg, Denmark

* so59@le.ac.uk

## Abstract

### Introduction

Plantar flexor weakness is an identified prospective factor for developing Achilles tendinopathy. Various authors have reported relationships between symptoms and weakness of this muscle group. Despite this relationship, many clinicians and researchers fail to examine Plantar flexor strength due to the cumbersome, stationary and expensive nature of an isokinetic dynamometer (IKD), known as the "Gold Standard". This study examined the validity and reliability of a fast, easy and portable device for assessing plantarflexion.

### Methods

Validity between the Cybex NORM® by Humac and the C-Station by Fysiometer was explored using Pearson correlation coefficient. Participants were randomly selected to start in the Cybex NORM® or the FysioMeter C-Station. Intra-rater reliability on the C-station was investigated by test-retest two days apart using Intraclass Correlation Coefficient (ICC). All testing involved isometric maximal force of the soleus muscle with the knee at 90 degrees flexion.

### Results

40 healthy university students were recruited for the validity part, while 65 healthy university students were recruited for the reliability part of the study. The mean peak torque on the IKD was 198.55Nm (SD 94.45) versus 1443.88 (412.82)N on the C-Station. The results of the Pearson correlation revealed an r-value of r = 0.72 with a 95%CI 0.52–0.84. The test re-test reliability was calculated as an ICC of 0.91 with a (95%CI 0.86–0.94).

### Conclusions

The C-Station by Fysiometer appears to provide valid measures and have excellent reliability for Plantar flexor isometric strength. It would appear suitable for both clinical and research work.

**Data Availability Statement:** All relevant data are available at: https://doi.org/10.25392/leicester.data.21770783.

**Funding:** The Authors received no specific funding for this work.

**Competing interests:** Martin Gronbech Jorgensen (0000-0002-3189-644X) is an owner/director of the Fysiometer company which originates from Aalborg University Denmark. He was not involved in data collection or analysis and had no influence on the results section of the report. He contributed to the production of the manuscript and planned methodology. This does not alter our adherence to PLOS ONE policies on sharing data and materials.

# Introduction

The Plantar flexor muscle group comprise the triceps surae encompassing the Gastrocnemius (medial and lateral heads) and Soleus, alongside the deep compartment muscles including Tibialis posterior, Peroneus Longus, PeroneusBrevis and the toe flexors. This muscle group as a whole has received much interest as weakness has been linked to numerous lower limb disorders such as tendinopathies of Achilles tendon, Achilles tendon rupture, calf muscle injuries (including "tennis leg"), medial tibial stress syndrome, tibial stress fractures, anterior cruciate ligament injuries, frailty and falls risk [1–4]. For instance, recent research has highlighted the association between low Plantar flexor strength and Achilles tendinopathy patients [3, 5–11], whilst other studies have shown the importance of Plantar flexor function to patient-reported outcomes post Achilles tendon rupture [12–14]. The existing literature reports an association between these injuries and reduced Plantar flexor power, with some prospective studies reporting the existence of deficits pre-injury [3], supporting the notion that Plantar flexor weakness may increase the risk of certain lower limb disorders [3, 15, 16].

The majority of the aforementioned literature has utilised an isokinetic dynamometer (IKD), which is considered the most accurate and reliable tool for measuring Plantar flexor strength. However, IKD's are very expensive, large and not portable. Furthermore, it requires specialist training with regular use and a significant time to complete testing. For instance, a recent protocol we developed took around 45–60 mins per patient [11, 17]. These practical limitations have led to poor uptake of IKD within clinical care and the inability of busy clinicians to quantify appropriate targets for rehabilitation. Research has likewise suffered, as many studies have failed to examine this important muscle group due to limited access to appropriate tools. It is obvious that a quicker and cheaper alternative needs to be developed.

Over the last few years, Nintendo Wii boards have been utilised via third party software (FysioMeter, Denmark) to measure various types of muscle strength, balance and reaction time. With a novel device (C-station) from FysioMeter, it is possible to complete Plantar flexor strength testing of the soleus muscle, alongside some of the deeper plantar flexors. This device was used for recent studies investigating therapeutic interventions [18], but the reliability and validity of this methodology are unknown. Without specific information on the psychometric properties of the device/test clinicians and researchers do not know if the device is appropriate for use.

## Aim

1. To determine the concurrent validity between the novel device a C-station (FysioMeter, Denmark) and an IKD (Humac Norm, CSMI solutions, USA)

2. To determine the intra-rater reliability of the novel device a C-station (FysioMeter, Denmark) when compared with dynamometry for measuring Plantar flexor isometric strength

# Method

## Study design

The study used two specific components to achieve the aims. The concurrent validity was examined by a randomised cross-over design to test the concurrent validity between the two devices. The participants were randomly allocated to either begin with the Cybex NORM® or the C-Station to reduce any order effect using a block randomisation procedure.

The reliability of the C-Station was tested using a test- re-test method, with testing being two days apart and by a single tester (intra-rater). The follow-up tests were completed at the same

time of day as the original testing to minimise the influence of the circadian rhythm [19]. The study followed the Guidelines for Reporting Reliability and Agreement Studies (GRRAS) [20] and met the consensus based standards for the selection of health measurements instruments (COSMIN) sample size recommendations of at least 50 participants for a good sample size.

## Sample size calculation

In order to conform with the Consensus Based Standards For Selection Of Health Measurement Instruments (COSMIN) recommendations we decided to ensure a minimum recruitment of 50 participants and allow for drop outs [21]. A sample size calculation was difficult to complete due to the limited data available. Due to this a pilot study of 20 subjects was completed and the ICC values from this population where used to develop a formal Sample size calculation. Based on a minimum acceptability of a 0.6 ICC value and an expected ICC value of 0.8, using an Alpha value of 0.05 and a power of 80% with a 10% drop out a total of 55 subjects would be needed.

## Study population

The study recruited a convenient sample of 40 university students for the concurrent validity component. All participants were healthy students without lower limb disorders (vascular, neurological or musculoskeletal). All consenting participants attended a university laboratory housing the Cybex NORMⓇ and the Fysiometer C-Station. Blinding participants regarding test method and test order was not possible, but the researcher and data analyst were blinded to results until the completion of statistical testing.

A further 25 participants were recruited from the university student population (total n = 65) for the test re-test reliability component of the study with no drop outs. This was an over recruitment. Inclusion criteria were the same as described above for the validity part. The increased number was used to improve the robustness of the reliability data and ensure we adhered to the COSMIN recommendations [21]. All participants were required to be injury free and fit and well at the time of testing and over 18 years of age.

## Ethic clearance

Ethical approval was sought and granted from the University ethics committee prior to study commencement (SON-8657-15). Written and verbal consent was given by all participants after reading the participant information sheet. All subjects had the ability to withdraw consent at any time point. Consent was taken by a trained researcher and witnessed by the research assistant and adhered to GCP standards [22]. All documentation was stored in keeping with the data protection act and research governance of the host institute.

## Plantar flexor testing

All testing involved isometric maximal force with the knee at 90 degrees flexion. Knee flexion around 90 degrees inhibits Gastrocnemius and biases testing to the Soleus muscle [23–25]. Isolating the Soleus muscles is important, as our previous work has highlighted the importance of the Soleus in diseases of interest, i.e. Achilles tendinopathy [11]. Testing was completed with the ankle in plantargrade as previous work has shown this position to be suitable. Joint angles were measured using a goniometer. The tests were conducted on the right leg for all participants, as previous work showed only slight variation between limbs [11], and this allowed for a more rapid test of participants. Participants were instructed to wear footwear in which they would attend a gym.

The isometric protocols were identical for each unit. A standardised test protocol, previously developed through extensive pilot testing of healthy controls and injured subjects, was used. The protocol consisted of a familiarisation component with three submaximal contractions followed by the testing component with three maximal contractions separated by a 30-second rest. Our extensive pilot work identified that 97% of subjects generated peak force within three or fewer repetitions. Moreover, the force differed by <3% in the remaining participants, who developed peak force outside the initial three repetitions. The 30 second rest period was also shown to be sufficient, with longer periods producing no increase in force output. There was a 10-minute rest period for subjects completing the cross-over concurrent validity study. All participants could see their real-time contraction data, which has previously been shown to influence output [26]. All participants were given standardised verbal encouragement of using identical intonation and verbal tone; this was done in an attempt to maintain participant effort/motivation on different test days.

## Dynamometer Isometric protocol for Plantar flexor testing

A Cybex NORM® IKD (Humac Norm, CSMI solutions, USA) was utilised using a protocol that was previously tested for reliability with test-retest ICC values of 0.9 [27]. All tests were completed on the dynamometer following the manufacturer's guidelines regarding positioning for flexed knee testing. Subjects were positioned in crook lying with the knee and ankle positioned following the defined protocol.

## Fysiometer isometric Plantar flexor testing

All measurements were completed using the FysioMeter C-Station system hardware and software (FysioMeter ApS, Denmark). The C-Station system comprises an aluminium case that holds a Nintendo Wii balance board (WBB) (Nintendo, Kyoto, Japan). The WBB is a small rectangular force plate, fitted with four uni-axial strain gauge transducers positioned in each corner. Data is transferred to a personal computer wirelessly via a Bluetooth device and analysed by the FysioMeter® software (version 1.5.0). The FysioMeter® software (Broenderslev, Denmark) analyses data simultaneously from all transducers at 100 Hz, with a 4th order Butterworth filter (cut-off 20 Hz). The device has an accuracy of 0.98N whilst measuring from 0-2943N.

(Fig 1 shows the C station and WBB set up).

The WBB is clipped into an aluminium frame to prevent transverse movement of the board. The aluminium frame has an adjustable belt system that allows the knee to be fixed. The use of the tensioner belts allows appropriate positioning of the ankle joint for Plantar flexor testing, and prevents knee lifting and moving away from the WBB. Testing was completed with participants seated on an examination couch/plinth and their foot on the Fysiometer platform with the ankle at plantargrade and the knee at 90 degrees of flexion. The metatarsal heads were positioned in the middle of the board (Identified by a horizontal line), and the foot was centred on a vertical line. Then the lower leg was weighed, and the belt tension was doubled (i.e. if the lower leg was 98.06N, the target tension was set to 198.13N). The fixation belt was fixated on the knee to hold the leg and allow for an isometric Plantar flexor contraction. This was followed by calibration to zero by the C-Station system before testing could begin.

## Data analysis

Data were recorded for each of the three maximal repetitions, and the peak reading was used for analysis. For the IKD testing, the values were Newton Metres (torque), and for the

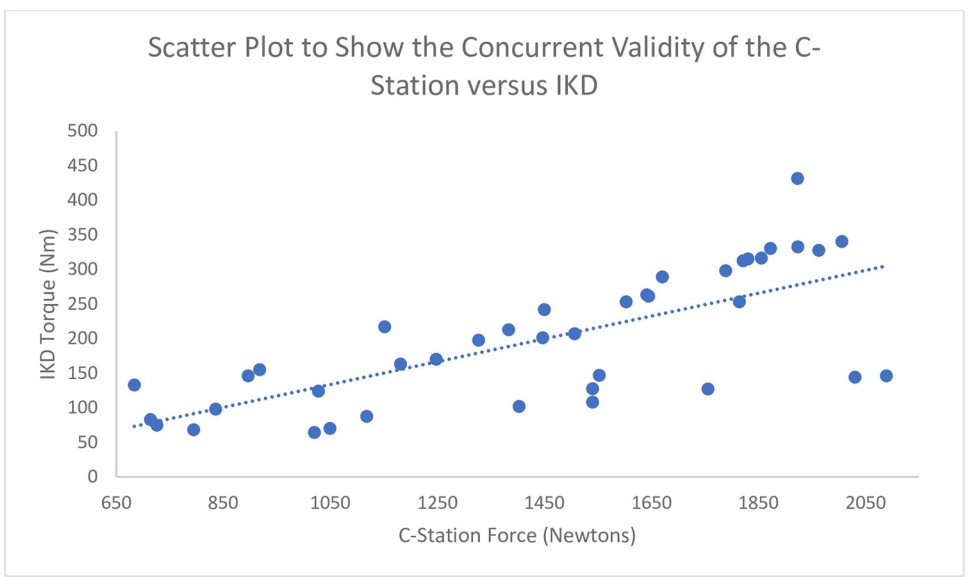

**Fig 1. A scatter plot showing the relationship between the C-Station measure and IKD measure with a line of best fit.**

Fysiometer the values were Newtons. The peak reading from the three maximal efforts was used for analysis as it is the value clinicians are interested in during clinical care and previous studies suggest the peak reading to be the most reliable measure [17]. All data were recorded in the native software (CSMI or FysioMeter) and transcribed into Microsoft excel. Cross-checking was employed for all data entries by a third research assistant to reduce input error. Once data collection was complete, the Microsoft Excel spreadsheet was imported into SPSS version 26 (SPSS Inc., Chicago, IL., USA.), where all analysed were conducted. Initially, data were analysed for simple descriptive statistics with the mean and standard deviation reported in Table 1. The data was then analysed for normality of distribution using Shaparo-Wilks test, all data was normally distributed. The concurrent validity was tested using a Pearson correlation coefficient comparing the Fysiometer data with the IKD (gold standard). The data was then plotted into a scatterplot with line of best fit. The test re-test reliability was analysed using intra-class correlation coefficients using a two-way mixed effects model and absolute agreement as recommended by Koo and Li (2016) [28] A good ICC level was evaluated as >0.7 according to accepted norms [28]. Standard error of the measurement was calculated using the ICC scores and SD. The Minimal Detectable Change (MDC) was also calculated using the formulae MDC95 = SEM × 1.96 ×√2. (1). The MDC represents the amplitude of change required to exceed the measurement error between the two measures. Knowledge of the MDC is essential for clinical interpretation and for later studies to determine if any change in these measures could be attributed to an intervention or injury, rather than error of the

**Table 1. Results of the test re-test using the C-Station.**

| | 1st test Mean (SD) | 2nd test Mean (SD) | Difference Mean (SD) | ICC (95%CI) | SEM (Newtons) | MDC (Newtons) |
|---|---|---|---|---|---|---|
| **Isometric plantarflexor force (Newtons)** | 1275.89 (398.50) | 1562.50 (389.09) | 13.38 (169.10) | 0.91 (0.86–0.94) | 52.79 | 146 |

ICC = intraclass correlation coeffiecient, Sem = Standard Error of the Measurement, MDC = Minimal detectable Change.

measurement. A Bland Altman plot (limits of Agreement) was also completed for analysis of the difference between the test re-test component of the study.

## Results

### Concurrent validity

The concurrent validity analysis was completed on 40 subjects using a Pearson correlation coefficient to compare the results of the C-station with the dynamometry unit, r-value of r = 0.72 (95%CI 0.52–0.84) with a significance value of <0.01 and is demonstrated in the scatter plot (Fig 1).

These 40 subjects had a mean age of 21.8 years (SD ± 4), weight of 81.9kg ± (SD 16.0) and height 174.8cm (SD ±10.3). The mean force was 1443.87N ± (SD 412.83) Newtons for the C-Station and 198.55 (± 94.45Nm) for the IKD (Humac Norm, CSMI solutions, USA).

### Test re-test reliability

Subjects in the test re-test reliability (n = 65) component had a mean age of 22.9 years ±4.3, weight of 87.2kg ± 15.6 and height 177.4cm ±9.5. 28 of the cohort were female (43%) The mean score for the 1st test was 1575.89 17N ± 398.50. The mean for the 2nd test was 1562.50N ± 389.09. The mean difference in scores of 13.38 (169.1) Newtons with the Minimal detectable change of 146 Newtons.

The ICC results were 0.91(95%CI 0.86–0.94) suggesting an excellent level of test re-test reliability (Table 1). Bland-Altman bias for absolute values was 1.36, LOA were -32.68 to 35.41 (Fig 2).

Analysis of the bland altman plot (levels of agreement) (Fig 2) shows that the Fysiometer C-station has high levels of agreement between the two test points. The discrepancy between the

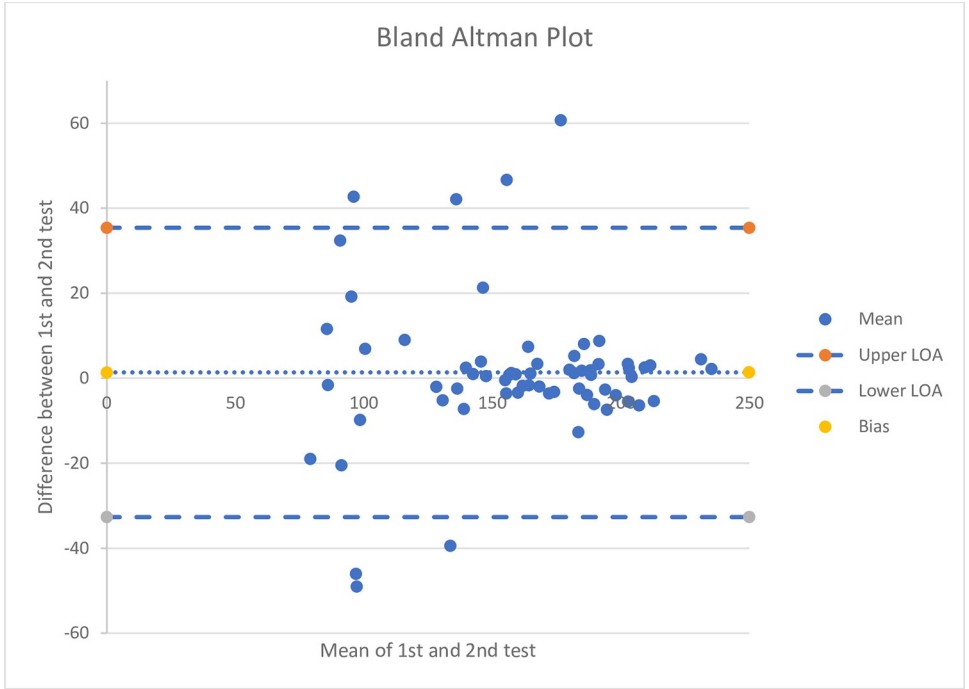

**Fig 2. A bland Altman plot for the test re-test reliability showing the limits of agreement, mean and bias levels.**

two test results is small and clinically insignificant. As the average increases there seems to be little different in variation.

## Discussion

This study is the first to examine the novel use of a Fysiometer for testing isometric Plantar flexor strength. The study showed a good correlation in plantarflexion strength measurements between the novel Fysiometer C station and an isokinetic dynamometer (IKD). The high level of correlation suggests that plantarflexion strength measured using the FysioMeter C-Station has a good level of concurrent validity. In conjunction with the high level of validity, the study also showed an excellent test re-test reliability, with a relatively tight confidence interval for the FysioMeter C-Station. This can be observed in the bland altman plots and clearly shows that the test has good absolute reliability in a large number of individuals.

There is a considerable need for a practical test of Plantar flexor peak force as the current options lie with IKD, which has significant limitations due to the expense and size of the units, as well as the time required for testing and the specialist training necessary. In contrast, the C station is substantially cheaper, more portable, quicker, and easier to use, which heightens the clinical feasibility of plantarflexion strength measurement. For instance, this study's IKD and WBB protocol had a 27-min difference in favour of the WBB test. The IKD took around 30 minutes for each subject to complete, while the FysioMeter C-Station took around 3 minutes. This makes it possible for clinicians to test patients within busy clinics and provide quantifiable data on strength during rehabilitation. The speed also makes it possible for large populations to be sampled within research which was not possible previously.

The importance of the Plantar flexors force-generating capacity to diverse clinical areas ranging from falls in elderly patients to sporting injuries and recovery post Achilles tendon rupture, highlights the requirement for a simple clinical test that can be replicated easily [1, 4–6, 8, 10, 29–31]. Therefore, the development of this test protocol should help both clinicians and researchers understand neuromuscular deficits associated with the many disorders that impact Plantar flexor function or occur as a result of Plantar flexor deficits. The use of Plantar flexor force as an outcome measure is noteworthy, particularly for Achilles tendon ruptures, where improved Plantar flexor force is associated with better patient-reported outcomes [29]. And for Achilles tendinopathy where the focus of treatment is often strength development [32, 33].

### Comparison with reliability of IKD

IKD Plantar flexor testing has received much interest over a long period of time [3, 18, 30, 34–36]. Despite this frequency there are few clinical studies utlising this measurement tool and those that do frequently use different/novel test parameters i.e. speeds/angular velocities, contraction modes (Conc:Conc, Eccentric:Eccentric, or Concentric:Eccentric), ankle ROMs, and knee angles [17]. Or report different metrics e.g. Total work done, average rep, best rep, endurance ratio, or fatigue index. All of these variations are often used without any awareness of the reliability of the chosen method or parameter. This large variation in chosen test methodology makes it very difficult to replicate studies or compare clinical cohorts. This is especially important when some IKD units do not allow for the angular velocity (test speed) to be modified beyond the pre-sets, thereby making comparison impossible. These issues subsequently led to this project.

The test re-test reliability of current IKD protocols is variable with values between 0.37 and 0.98 [5, 17, 34, 35, 37–42]. Higher reliability is only normally achieved in elite or semi-elite athletes and studies on a single gender which fail to replicate clinical cohorts. Mixed gender cohorts with a more varied age tend to give larger variation in strength data and subsequently

variable ICC's, but have more clinical relevance [17]. The results of this project show that the Fysiometer C-Station system has excellent absolute test re-test reliability (ICC 0.91), greater simplicity and less potential methodological variation than IKD testing. In fact the achieved ICC suggests that the Fysiometer is near to the top levels of reliability achieved with IKD testing and should be used with confidence [3, 17, 35].

## Clinical application

Due to the level of portability, cost, and simplicity, speed of testing, we recommend using the Fysiometer C-Station in clinical, research and sporting settings to monitor Plantar flexor strength. We acknowledge the extra depth of information that can be gathered from IKD like the position peak force is obtained in, the force trace data, components of the stretch-shorten cycle (tissue compliance), muscle function under different contraction modes and angular velocities. But there are many methodological variations inherent in IKD use that impact the reliability of measurements, i.e. tests using angular velocities that are not functional, contraction modes not representing normal movement (eccentric infrequently measured), positions used for testing and most importantly the speed/time and cost of these devices. We suggest Isometric testing should be the first line of clinical or research testing where large numbers of people need to be testing quickly. But where further more detailed understanding of the muscles function is required, then IKD measurement or force plate testing may be incorporated. This is particularly true if the role of energy storage (stretch shorten cycle) in passive connective tissue is required.

The clinical and research limitations of this isometric protocol are the inability to assess the muscles function across various modes: concentric, eccentric or stretch-shorten cycle, and various contraction speeds. The test position (ankle position) can obviously be altered dependent on relevance, but that can be hard to discern in the first place and difficult to standardise. Hence this test utilised a fixed position–plantargrade as it is easier to standardise.

## Limitations

The MDC found within this study, 146 Newtons, is moderate, suggesting alterations above this level would represent an actual change in the measure or between-group difference. However, the population being sampled must be considered. The population in this study represented a university cohort of students of mixed gender. Due to this, there is a large variance in Plantar flexor strength which directly influences the MDC. Using a more specific population i.e., elite footballers of the same gender, would give less variation and therefore produce a smaller MDC. However, this study represents typical mixed gender studies in clinical and sporting populations and offers excellent clinical utility. Therefore the MDC identified is very conservative.

The cross-over design and sample size improve the confidence and generalizability of the findings. It would have been preferable to have had the same size cohort for both the validity and reliability arms of the study; however, the time constraints and laboratory access prevented this. The current study participants are regular, healthy, younger adults; thus, the results can only be generalised to this group. Additional studies should be conducted on other populations, and further work is needed to ascertain population norms across various age- and patient groups and different sporting backgrounds. This data will help ascertain targets for rehabilitation and may serve as a marker in prospective epidemiology research.

The use of the flexed knee test position acts to inhibit Gastrocnemius force generation whilst still working Soleus maximally [24, 25, 43, 44]. This is of some importance, as Soleus has been identified by numerous researchers as the main locomotive muscle during walking and

running generating forces of around 8x bodyweight compared with the smaller gastrocnemius contributions, around 3x body weight [45–48]. And the most commonly injured of the plantarflexors [49–51]. The limitation to flexed knee testing warrants some consideration and it would probably be useful to develop a rig for testing in knee extension so that the Gastrocnemius' contribution can also be included [24, 25, 43]. However our previous work has identified similar force generation irrespective of knee flexion angle in runners with and without Achilles tendinopathy [11], suggesting that this may produce little additional data. Other studies have however identify extended knee positions produce greater forces, but this may be population specific, i.e. runners versus non runners [44].

As previously identified the new device provides a robust, speedy, low cost method for testing plantar flexor strength with excellent portability and few limitations.

## Conclusion

Using the Fysiometer C-Station to test plantar flexor strength appears to be a highly valid and reliable test. The psychometric test results suggest this protocol is suitable for both clinical and research work. This test protocol will improve the speed and costs associated with measuring large numbers of individuals. Due to these facts we recommend this test protocol and the Fysiometer C-station be used in clinical and research work.

## Author Contributions

**Conceptualization:** Seth O'Neill, Alice Weeks, Jens Eg Nørgaard, Martin Gronbech Jorgensen.

**Data curation:** Seth O'Neill, Alice Weeks.

**Formal analysis:** Seth O'Neill.

**Investigation:** Seth O'Neill, Alice Weeks.

**Methodology:** Seth O'Neill, Alice Weeks, Martin Gronbech Jorgensen.

**Project administration:** Seth O'Neill.

**Resources:** Seth O'Neill.

**Software:** Seth O'Neill.

**Supervision:** Seth O'Neill.

**Visualization:** Seth O'Neill.

**Writing – original draft:** Seth O'Neill.

**Writing – review & editing:** Seth O'Neill, Jens Eg Nørgaard, Martin Gronbech Jorgensen.

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
