## [Decision Letter · Decision Letter 0]

9 Nov 2022

PONE-D-22-19672Validity And Intrarater Reliability Of A Novel Device For Assessing Plantar flexor StrengthPLOS ONE

Dear Dr. O'Neill,

Thank you for submitting your manuscript to PLOS ONE. After careful consideration, we feel that it has merit but does not fully meet PLOS ONE’s publication criteria as it currently stands. Therefore, we invite you to submit a revised version of the manuscript that addresses the points raised during the review process.

We look forward to receiving your revised manuscript.

Kind regards,

Jongsang Son, Ph.D.

Academic Editor

PLOS ONE

“Martin Gronbech Jorgensen (0000-0002-3189-644X) is an owner/director of the Fysiometer company which originates from Aalborg University Denmark. He was not involved in data collection or analysis and had no influence on the results section of the report. He contributed to the production of the manuscript and planned methodology.”

5. Please include your tables as part of your main manuscript and remove the individual files. Please note that supplementary tables (should remain/ be uploaded) as separate ""supporting information"" files.

Reviewers' comments:

Reviewer's Responses to Questions

**Comments to the Author**

1. Is the manuscript technically sound, and do the data support the conclusions?

Reviewer #1: Yes

Reviewer #2: Yes

2. Has the statistical analysis been performed appropriately and rigorously? 

Reviewer #1: Yes

Reviewer #2: Yes

3. Have the authors made all data underlying the findings in their manuscript fully available?

Reviewer #1: Yes

Reviewer #2: No

4. Is the manuscript presented in an intelligible fashion and written in standard English?

Reviewer #1: Yes

Reviewer #2: No

5. Review Comments to the Author

Reviewer #1: This is a study on the reliability and validity of a cost-effective muscle strength test device. Information on accuracy, resolution, reliability, and validity is essential in clinical practice.

In this study, the new device(hardware) whose reliability needs to be investigated is the Nintendo Wii balance board (NWBB). The software - Fysiometer – does not seem to be critical. However, NWBB has already been studied in many previous studies. The differences from previous studies seemed to be clearly stated.

Most previous NWBB studies investigated the inter-rater, intra-rater reliability, and concurrent validity of over 60 patients. It seems necessary to gather more subjects, and statistical methods should be reconsidered by referring to the latest similar studies.

In addition, the clinical applicability of the new method needs to be emphasized by highlighting its strength despite the limitation of the new device.

The references are old. Please add the latest studies.

Reviewer #2: Minor comments:

- Line 67-68: Add please reference to the part "(...) supporting the notion that Plantar flexor weakness may increase the risk of certain lower limb disorders.

- Methods: I would suggest combining Concurrent validity and Test Re-test intra-rater Reliability and creating a sub-paragraph as "Study design".

Major comments:

- Line 70: Please explain the term “Gold Standard”.

- Introduction: Please add a clear perspective of the research problem, and focus on the research explanation about the novelty of this research.

- Introduction: Please add a hypothesis.

- Methods: An estimation of sample size and test power would be appreciated.

- Data analysis: Please identify the exact model and type of ICC according to Koo and Li (2016).

- Results: This section should be re-arranged and focused on the exact results, including absolute and relative reliability. Currently, the section is very unclear to the reader.

- Discussion: A major comment - this section is very pour according to references and literature.

- Limitations: Please re-arrange this part. Why the limitations states one page? It may indicate a huge weakness of this paper. Add the rest of this part into the discussion and prepare clear findings to discuss further.

6. PLOS authors have the option to publish the peer review history of their article (what does this mean?). If published, this will include your full peer review and any attached files.

Reviewer #1: No

Reviewer #2: No

---

## [Author Response · Author response to Decision Letter 0]

3 Jan 2023

PLOSONE review Comments and responses

We thank the reviewers for their comments and expertise. We have individually responded to each comment giving a response and action when and where relevant. The text has subsequently been modified and additional elements are identified in yellow. 

Comment: Reviewer #1: This is a study on the reliability and validity of a cost-effective muscle strength test device. Information on accuracy, resolution, reliability, and validity is essential in clinical practice.

In this study, the new device(hardware) whose reliability needs to be investigated is the Nintendo Wii balance board (NWBB). 

Response: Regarding the hardware – this specifically includes the metal framed c station which is part of the Fysiometer system. This is critical to testing. Obviously this information was not well documented in our case so a picture has been added. 

Action: we have added a picture to help show the hardware and NWBB.

Comment: The software - Fysiometer – does not seem to be critical. However, NWBB has already been studied in many previous studies. The differences from previous studies seemed to be clearly stated.

Response: We would argue the Fysiometer software is critical to testing as otherwise you would need to use the native Nintendo software and set up as if you were measuring an individuals weight. This would make It difficult to record peak force. As mentioned the hardware from Fysiometer is crucial to the ability to test the plantarflexors. 

Action: A picture has been added to the manuscript. 

Comment : Most previous NWBB studies investigated the inter-rater, intra-rater reliability, and concurrent validity of over 60 patients. It seems necessary to gather more subjects, and statistical methods should be reconsidered by referring to the latest similar studies.

Response: We agree, the COSMIN guidelines for reporting reliability studies were not met and we have added to the initial reliability data. We have therefore updated the paper to include a further 25 subjects who completed testing with us for test re-test reliability. Taking the size to 65. Regarding statistical methods, we are keen to keep this manuscript relevant to clinicians and therefore only reported key stats – i.e. correlation, ICC, and MDC, alongside the Bland Altman tests. If you feel we should complete some other specific tests please let us know. 

Action : Manuscript modified to reflect this. But statistical tests kept the same except further information on the type of ICC completed as requested by reviewer 2. 

Comments: In addition, the clinical applicability of the new method needs to be emphasized by highlighting its strength despite the limitation of the new device.

Response: We do not feel the new device has any real limitations other than its inability to test different contraction modes, however the sections on clinical applicability and limitations reflect this. We have modified the clinical application section to reflect your comments more specifically and emphasize the strengths of the new device in the same section and strengths/limitations section. Please also note the overall conclusion. 

Actions: Modifications to the manuscript – line 262-267 and line 304-305 

Comments: The references are old. Please add the latest studies.

Response: We would disagree with this statement. The age of a reference does not render it obsolete. References do not need to be updated unless new studies are completed – they aren’t or haven’t been and only then when they find something new. It is best to use the seminal papers in the area. This is something we have done.

If there are specific references that are outdated please do highlight them but we are not aware of any specific to the clinical side or the device. 

Actions: None taken. But we are happy to amend if there are specific references identified. 

Reviewer #2: Minor comments:

Comments: - Line 67-68: Add please reference to the part "(...) supporting the notion that Plantar flexor weakness may increase the risk of certain lower limb disorders.

Response: The reference at the beginning of this sentence is relevant to this statement. However for clarity we have added several to this sentence so that the audience knows it is an accepted consideration. 

Action: supporting the notion that Plantar flexor weakness may increase the risk of certain lower limb disorders. (24,30)

Comments: - Methods: I would suggest combining Concurrent validity and Test Re-test intra-rater Reliability and creating a sub-paragraph as "Study design".

Response:

Action: We have done as suggested. Thank you for this. 

Comments: - Line 70: Please explain the term “Gold Standard”.

Response: This terminology is a typical English statement and as such needs removing and replacing. 

Action: Insertion of statement to say “which is considered the most accurate and reliable tool for measuring Plantar flexor strength”

Major comments:

Comments: - Introduction: Please add a clear perspective of the research problem, and focus on the research explanation about the novelty of this research.

Response – line 70 and 82 provides the relevant background about the problems with plantar flexor testing using IKD and leads into the development of the C-Station. 

Action : We have added additional commentary into the introduction line 69-70 and 81-84 add to the existing manuscript. 

Comment:- Introduction: Please add a hypothesis.

Response: We do not think a hypothesis is appropriate as we are examining the reliability and validity of the device. We report the study in such a way. It does not therefore warrant a hypothesis, especially as the results represent these constructs and do not test a hypothesis. The aims clearly identify the necessary information. 

Action: none given

Comment:- Methods: An estimation of sample size and test power would be appreciated.

Response: We previously had not followed the COSMIN recommendations for sample size and have therefore modified the sample size with a further 25 subjects taking the reliability element to 65 subjects. 

We have also completed an appropriate sample size calculation for an ICC/reliability study.. 

Action: The text has been modified to reflect the increase in sample size and a section in the method on sample size has been added. 

Comment: - Data analysis: Please identify the exact model and type of ICC according to Koo and Li (2016).

Response: https://pubmed.ncbi.nlm.nih.gov/27330520/ We followed the advice from the reviewer and have used the papers recommendations to form the ICC method and wording within the manuscript. 

Action: Modified test (highlighted in yellow) line 189-191

Comments: - Results: This section should be re-arranged and focused on the exact results, including absolute and relative reliability. Currently, the section is very unclear to the reader.

Response: 

Action: We have modified the results section using sub-headings to help guide the reader and modified the text. And ensured the ICC information was prominent. We feel with the addition of the tables in to the final manuscript that this section will read much more clearly. 

Comments: - Discussion: A major comment - this section is very pour according to references and literature.

Response – It is unclear what is pour exactly in relation to the references and literature. We assume this is the lack of discussion in relation to other literature however since this is a novel test and there is no other published data we would need to discuss IKD literature. This would lose the key focus of the paper – i.e. reliability of the C station. 

Action: Modifications to the Discussion . We have added more references to some statements throughout the discussion. 

Comments: - Limitations: Please re-arrange this part. 

Comments: Why the limitations states one page? It may indicate a huge weakness of this paper. Add the rest of this part into the discussion and prepare clear findings to discuss further.

Response: We have tried to make the limitations clear and consider what this really means in the real world by helping the reader understand the population sampled, the study design, and the positional limitations. 

The size of the limitations section shows our thorough awareness of limitations rather than the condensed version requested. We feel this is better for the reader rather than limiting this discussion. Our concern with moving much of this into the discussion more generally would hinder the readers understanding and hide these issues. 

We would be happy to take an editorial decision on this. 

Action: the Discussion and limitations section has been altered 

.

---

## [Decision Letter · Decision Letter 1]

15 Feb 2023

Validity And Intrarater Reliability Of A Novel Device For Assessing Plantar flexor Strength

PONE-D-22-19672R1

Dear Dr. O'Neill,

We’re pleased to inform you that your manuscript has been judged scientifically suitable for publication and will be formally accepted for publication once it meets all outstanding technical requirements.

Kind regards,

Jongsang Son, Ph.D.

Academic Editor

PLOS ONE

Additional Editor Comments (optional):

Reviewers' comments:

Reviewer's Responses to Questions

**Comments to the Author**

1. If the authors have adequately addressed your comments raised in a previous round of review and you feel that this manuscript is now acceptable for publication, you may indicate that here to bypass the “Comments to the Author” section, enter your conflict of interest statement in the “Confidential to Editor” section, and submit your "Accept" recommendation.

Reviewer #1: All comments have been addressed

Reviewer #2: All comments have been addressed

2. Is the manuscript technically sound, and do the data support the conclusions?

Reviewer #1: Yes

Reviewer #2: No

3. Has the statistical analysis been performed appropriately and rigorously? 

Reviewer #1: Yes

Reviewer #2: Yes

4. Have the authors made all data underlying the findings in their manuscript fully available?

Reviewer #1: Yes

Reviewer #2: No

5. Is the manuscript presented in an intelligible fashion and written in standard English?

Reviewer #1: Yes

Reviewer #2: Yes

6. Review Comments to the Author

Reviewer #1: Comments raised in a previous review were all addressed by the authors.

The revised manuscript is described a technically sound piece of scientific research with data that supports the conclusions.

The manuscript presented is in an intelligible fashion and written in standard English

Reviewer #2: (No Response)

7. PLOS authors have the option to publish the peer review history of their article (what does this mean?). If published, this will include your full peer review and any attached files.

Reviewer #1: No

Reviewer #2: No

---

## [Editor Report · Acceptance letter]

21 Mar 2023

PONE-D-22-19672R1 

Validity And Intrarater Reliability Of A Novel Device For Assessing Plantar flexor Strength 

Dear Dr. O'Neill:

I'm pleased to inform you that your manuscript has been deemed suitable for publication in PLOS ONE. Congratulations! Your manuscript is now with our production department. 

Kind regards, 

on behalf of

Dr. Jongsang Son 

Academic Editor

PLOS ONE